# Management of Bladder Cancer Patients with Clinical Evidence of Lymph Node Invasion (cN+)

**DOI:** 10.3390/cancers14215286

**Published:** 2022-10-27

**Authors:** Bartosz Małkiewicz, Adam Gurwin, Jakub Karwacki, Krystian Nagi, Klaudia Knecht-Gurwin, Krzysztof Hober, Magdalena Łyko, Kamil Kowalczyk, Wojciech Krajewski, Anna Kołodziej, Tomasz Szydełko

**Affiliations:** 1Department of Minimally Invasive and Robotic Urology, University Center of Excellence in Urology, Wroclaw Medical University, 50-556 Wroclaw, Poland; 2Department of Dermatology, Venereology and Allergology, Wroclaw Medical University, 50-368 Wroclaw, Poland

**Keywords:** bladder cancer, clinically positive lymph nodes, diagnosis, treatment, lymphadenectomy, immunotherapy

## Abstract

**Simple Summary:**

This review presents the current status of diagnostic and treatment options in bladder cancer (BCa) patients with clinically positive lymph nodes (cN+). There is no conclusive evidence regarding the management of cN+ patients, as most scientific associations do not distinguish the group in their guidelines or differ in the treatment options. A multimodal approach with a combination of neoadjuvant chemotherapy (NAC) and radical cystectomy (RC) with pelvic lymph node dissection (PLND) is associated with the best long-term survival in cN+ patients. In those patients, the extended template of PLND (ePLND) is recommended. Emerging evidence indicates that it is comparable to NAC results of adjuvant chemotherapy (AC); however, there is a lack of studies focusing on cN+ patients. The response to chemotherapy (ChT) is crucial for the prognosis of cN+ patients. Therefore, with a significant percentage of ChT-ineligible patients, immunotherapy has achieved growing importance in neoadjuvant and adjuvant treatment. Patients with cN+ BCa demand special attention, as the oncological outcomes are significantly worse for this group.

**Abstract:**

The purpose of this review is to present the current knowledge about the diagnostic and treatment options for bladder cancer (BCa) patients with clinically positive lymph nodes (cN+). This review shows compaction of CT and MRI performance in preoperative prediction of lymph node invasion (LNI) in BCa patients, along with other diagnostic methods. Most scientific societies do not distinguish cN+ patients in their guidelines; recommendations concern muscle-invasive bladder cancer (MIBC) and differ between associations. The curative treatment that provides the best long-term survival in cN+ patients is a multimodal approach, with a combination of neoadjuvant chemotherapy (NAC) and radical cystectomy (RC) with extended pelvic lymph node dissection (ePLND). The role of adjuvant chemotherapy (AC) remains uncertain; however, emerging evidence indicates comparable outcomes to NAC. Therefore, in cN+ patients who have not received NAC, AC should be implemented. The response to ChT is a crucial prognostic factor for cN+ patients. Recent studies demonstrated the growing importance of immunotherapy, especially in ChT-ineligible patients. Moreover, immunotherapy can be suitable as adjuvant therapy in selected cases. In cN+ patients, the extended template of PLND should be utilized, with the total resected node count being less important than the template. This review is intended to draw special attention to cN+ BCa patients, as the oncological outcomes are significantly worse for this group.

## 1. Introduction

Bladder cancer (BCa) is considered the most common urinary tract malignancy. The incidence is more significant in men than in women, and the highest rate is observed in Europe, reaching 36.7 per 100,000 in Spain. The highest mortality rate reaches 8.4 per 100,000 in Eastern Europe [1,2]. The management of the lymph nodes (LNs) requires insightful reflection, as, apart from the local stage, nodal metastases are the most significant prognostic factor in BCa patients. Additionally, the current guidelines regarding clinically-positive lymph nodes (cN+) patients are imprecise, differing in recommendations. The presence of LN metastasis in patients with muscle-invasive bladder cancer (MIBC) is associated with a worse prognosis, and each additional positive LN in the range of 1 to 4–6 is associated with lower survival rates [3,4,5]. The 5-year overall survival (OS) in node-positive bladder cancer (N+ BCa) was established at 30–32% in patients receiving treatment, while in patients without lymphatic spread it reaches up to 85% in neoadjuvant chemotherapy (NAC) responders [6]. N+ BCa remains conceivably curable prior to systemic metastasis [5,7,8,9]. The standard of treatment in MIBC involves radical cystectomy (RC) with bilateral pelvic lymph node dissection (PLND). Moreover, in eligible patients, NAC is advisable [10]. Due to occult metastasis, relapses after surgery are observed [11]. Considering the survival rates and the number of relapses after surgery, additional or novel treatment options are needed. However, the debate is still ongoing regarding whether chemotherapy (ChT) and radiation therapy (RT) can improve survival [12,13,14]. Diagnosis of N+ patients might be challenging, as in around 25% of patients, lymph node involvement (LNI) may not be noticeable at the time of imaging [15]. As patients with N+ status have been linked with worse oncological outcomes, this review summarizes the current knowledge about diagnostic and treatment options in cN+ BCa.

## 2. Data Acquisition 

For the purposes of this non-systematic review, we conducted a comprehensive search of the English language literature for original articles, meta-analyses and reviews using PubMed and grey literature through June and August 2022. We searched for various combinations of the following terms: bladder cancer, clinically positive lymph nodes, diagnosis, treatment, lymphadenectomy, and immunotherapy. We found 411 related articles, and the final number of papers selected for this manuscript was 286. Studies with the highest level of evidence and relevance to the discussed topics (229) were selected, with the consensus of the authors. 

## 3. Diagnosis of Lymph Node Invasion (LNI)

Staging lymph node metastases is one of the three elements of the TNM classification system. While transurethral resection (TURB) is usually used to confirm the diagnosis of a suspected bladder tumor, additional imaging is required for staging, including detecting LN metastases [16]. The most prevalent techniques are computed tomography (CT) and magnetic resonance imaging (MRI), with 18F-fluorodeoxyglucose positron emission tomography (18F-FDG PET) being increasingly utilized in clinical practice, albeit still not considered a standard. As discussed below, most guidelines do not indicate which technique is better for detecting LN metastases. Nevertheless, contrast-enhanced CT remains, in theory and in practice, the mainstay of imaging used for BCa staging, being recommended as first-line imaging in nearly all guidelines of major urological and oncological societies [17]. 

According to the European Association of Urology (EAU) guidelines, both MRI and CT demonstrate similar, relatively low sensitivity and specificity in detecting LN metastases, emphasizing that the possibility of an assessment based solely on their performance is limited. In both of these imaging techniques, enlarged nodes should be considered pathological if the maximum short-axis diameter exceeds 8 mm for pelvic nodes and 10 mm for abdominal nodes. Overall, CT of the chest, abdomen, and pelvis, including some form of CT urography, is recommended as first-line imaging for staging [18,19]. The American Society of Clinical Oncology (ASCO) has not issued its own BCa guidelines, but has announced its endorsement of the guidelines from EAU [20]. The advice from the European Society for Medical Oncology (ESMO) is very similar to those of the EAU, postulating similar results of CT and MRI in detecting LN metastases. The dimensions of the LN requiring attention are also the same. No clear recommendation is given with regard to what CT or MRI should be utilized as first-line imaging in staging. However, it is recommended to choose MRI when an accurate determination of the depth of invasion is needed, due to its higher accuracy [21]. The guidelines issued jointly by the American Urological Association (AUA), the American Society for Radiation Oncology (ASTRO), and the Society of Urologic Oncology (SUO) do not discuss the diagnosis of LNI in BCa. A CT scan with contrast of the pelvis and abdomen as well as an X-ray/CT of the chest are recommended for staging. MRI should be counseled if a CT cannot be performed [22]. The joint Société Internationale d’Urologie (SIU) and the International Consultation on Urological Diseases (ICUD) guidelines state that CT and MRI are equivalent in detecting the metastatic LNs. They point out that the lack of well-established criteria to distinguish between malignant and benign LNs is a significant limit in the successful detection of metastases in normal-sized nodes. They also mention the promising results of lymphotropic nanoparticle-enhanced MRI in the detection of micrometastases in normal-sized lymph nodes (with a sensitivity of up to 96%), listing the small amount of research and lack of studies on its impact on patient management as the major obstacles of wider usage. Overall, a CT scan with contrast of the abdomen and pelvis, including an excretory phase study, is recommended to investigate nodal and distant metastases in patients with BCa. MRI is advised only if CT contrast is not tolerated [23]. The National Comprehensive Cancer Network (NCCN) guidelines and the National Institute for Health and Care Excellence (NICE) guidelines both recommend CT or MRI, without specifying which is better, for staging in BCa [24]. If the findings in MRI or CT are indeterminate and the risk of metastatic disease is high, it is also recommended that the 18F-FDG PET be considered before the radical treatment [25].

The number of studies directly comparing CT to MRI in nodal staging of BCa is limited. The high patient population heterogeneity and the large variety of techniques used (contrast materials, protocols) make it difficult to compare the results of studies involving only one type of imaging. Despite significant advances in imaging technology, multiple reviews on this topic confirmed comparable, relatively low staging accuracy in both modalities [26,27,28,29]. Evidence of the superiority of either technique remains ambiguous— for instance, McKibben et al. indicate a clear predominance of MRI (with an accuracy of 54–97% and 73–98% for CT and MRI, respectively), while Bostrom et al. describe virtually identical results of both modalities (with an accuracy of 70–97% and 73–98% for CT and MRI, respectively) [30,31]. A recent meta-analysis by Woo et al. pooled 2928 patients from 23 studies, showing the combined sensitivity of MRI in the detection of metastatic LNs to be 56%, with a specificity of 94% [32]. Considering the similar results of both techniques, with some studies indicating the superiority of MRI, no need for ionizing contrast agents, and no radiation exposure, the use of MRI is encouraging [33,34]. Further research, particularly an RCT directly comparing the two techniques, would provide definitive evidence.

In most guidelines, as well as in clinical practice, the size of the LN is the main criterion for distinguishing between normal and suspicious LNs [35]. However, normal-sized nodes can be malignant; conversely, reactively enlarged ones may reveal no cancer deposits [36]. This is probably one of the primal causes of low accuracy in detecting LNI, mainly due to the relatively common presence of metastases in normal-sized LNs [17]. As a solution, additional criteria were proposed, such as LN shape, internal architecture, number of loco-regional LNs, and utilization of new contrast agents [37,38]. However, they have not gained widespread use so far.

The utilization of 18F-FDG PET/CT in the staging of BCa has been under consideration for many years [39,40,41]. Combining the anatomical information from CT with glucose metabolism (which is increased in metastatic LNs) is a widely accepted method in oncology [36]. However, as of today, no major guideline recommends its routine use. While some studies investigating the accuracy of 18F-FDG PET demonstrated promising results, others showed no significant improvement in diagnostic efficacy compared with conventional techniques [39,42,43,44,45,46,47,48,49]. The results of the selected studies are presented in Table 1. In order to improve the specificity and accuracy of 18F-FDG PET, alternative radiotracers, such as C11-Choline and C11-methionine, were proposed. A meta-analysis by Kim et al. pooled 282 patients from 10 studies that used C11-Choline and demonstrated a sensitivity of 66% (with a specificity of 89%) [50]. The data on C11-Choline is limited, but shows results comparable to conventional imaging techniques [51]. Overall, more research is needed to make a firm recommendation for the routine use of 18F-FDG PET.

Ultra-small-particle superparamagnetic iron oxide (USPIO) has been suggested as a possible alternative technique for detecting BCa LN metastases [62,63]. This method is based on the intravenous administration of iron oxide nanoparticles, which are then phagocyted by macrophages and taken up to LNs, where they remain for a few days. This uptake is reduced in malignant LNs, where healthy tissue is replaced with malignant cells. The superparamagnetic iron oxide can then be detected by T2 MRI. Due to the higher density of macrophages, benign LNs have higher signal intensity compared to malignant ones [30]. Several studies have reported encouraging results with excellent accuracy in the detection of metastatic LNs [64]. However, due to the complex, time-consuming, expensive interpretation procedure, which requires expertise, USPIO will be utilized in clinical trials and selected cases, but its usage is unlikely to become standard practice [31,65].

Due to the low accuracy of traditional methods in the staging of BCa, various risk-stratification models and nomograms were designed to improve it. The first attempts to predict LNI in BCa patients by Karakiewicz et al. in 2006 showed promising results; however, in independent studies, the accuracy turned out to be significantly limited [66,67,68]. In 2012, two small, single-center designs demonstrated better results, but lacked prospective validation [69,70]. Attractive nomogram designs based on more niche parameters, such as a genomic-clinicopathologic nomogram by Wu et al., which combines clinicopathological parameters with an LN-status-related mRNAs classifier, have also been developed [71]. However, due to the difficulty of performing genomic tests outside of research institutions, the chances of their popularization in the future are low. The first nomogram to reliably predict LNI in the patients treated with RC and extended pelvic lymph node dissection (ePLND) was recently developed by Moschini et al. It is based only on routinely available parameters (age, cT, cN, lymphovascular invasion, presence of carcinoma in situ) and has a prediction accuracy of 73%, which, according to the authors, could lead to avoidance of up to 12% lymphadenectomies at the cost of missing only 3% cN+ patients [72]. A similar nomogram, which was designed and tested on a much larger group of patients (10,653), demonstrated comparable accuracy and high reliability in predicting LNI [73]. Another design, created by Venkat et al. in 2021 and based on 6143 patients, achieved an even higher accuracy of 87.8%, significantly higher than most results in studies on imaging modalities [74]. The authors used following variables to predict LNI: age, race, sex, Charlson Comorbidity Index (CCI), cN, cT, grade, lymphovascular invasion, surgical margin, and tumor size. As these tools are already available and have been proven effective, their wider adoption combined with other prognostic factors and imaging could lead to better detection rates and, consequently, better treatment outcomes. Further research into clinical application and its impact on patient management is required.

## 4. Treatment of cN+ Patients

### 4.1. Guidelines

Current guidelines remain inconsistent in establishing proper management strategies for N+ patients, both for cN+ and pN+ subgroups. Table 2 presents the summary of the most pivotal recommendations from popular guidelines.

EAU guidelines highlight the low sensitivity and specificity of CT and MRI in nodal staging. This may be why the guidelines do not provide information on cN+ management. Nevertheless, the EAU proposes three major management options for N+ patients: RC with NAC or AC, radical chemoradiotherapy (ChRT), and immunotherapy with nivolumab. The guidelines further emphasize that the benefits of AC are still under debate. An immunotherapeutic approach with nivolumab is advised only for selected pT3/4 and/or pN+ patients. cN+ patients are not included in this recommendation [19].

AUA guidelines do not distinguish between different approaches for cN1 and pN1 groups. However, they recommend that N1 patients should receive RC with cisplatin-based NAC. Patients who have not received cisplatin-based NAC and have non-organ confined disease (pT3/4 and/or N1) should be offered AC [22].

ESMO guidelines indicate that cN1 patients should receive surgical treatment, but it should be considered whether to institute NAC or not [21].

NCCN guidelines advise five primary therapeutic pathways for N1 patients. These include NAC followed by RC or RC alone for chemotherapy-disqualified patients, bladder preservation with concurrent ChRT, ChRT alone, and RT. The guidelines indicate that cN1 patients have better outcomes when RC is preceded by NAC. For cN2-3 patients with stage IIIB disease, the guidelines advise either downstaging systemic therapy or concurrent ChRT [75].

The National Institute for Health and Care Excellence (NICE) guidelines do not differentiate between cN+ and pN+ patients. They propose two therapeutic options: RC with NAC and RC with AC. The latter should be a primary therapeutic option for patients for whom NAC is unsuitable [25]. 

### 4.2. Clinical Evidence, Surgery

As no conclusive evidence regarding cN+ management exists in the guidelines, discussing this common clinical situation is mandatory. Since cN+ BCa is generally considered in the same context as a metastatic disease (despite the local stage), multiple studies have investigated the outcomes of different treatments for those patients [9,76,77,78,79]. Patients with cN+ are generally considered for systemic induction chemotherapy (IC) [80]. The researchers found NAC followed by RC as the curative treatment with the best long-term survival, particularly in patients with a good response to NAC. Several researchers reported encouraging outcomes in extending the treatment to multimodal therapy, demonstrating OS improvement and even long-term survival in patients with initially unresectable BCa, who underwent IC with subsequent RC after a good response to IC [79,81]. Including radiotherapy in the treatment did not improve the survival of cN+ patients [82]. Patients should be eligible for curative treatment whenever possible, as palliative treatment is associated with a significantly worse survival rate [83]. However, due to impaired renal function, ChT might be challenging to perform in the elderly, which is one of the significant limitations of this treatment. The proportion of curative and palliative treatment decreases with age to less than 10% in octogenarians, which is an important issue, as the average age of BCa diagnosis is 73 years [84,85]. As the value of ChT in the therapeutic pathway of the cN+ patients is crucial, we specified this aspect further in the article and discussed the surgery details below. 

Many reasons may be utilized for the rationale behind post-IC RC. Firstly, although BCa is chemosensitive, IC is rarely curative [86]. Secondly, RC is the best possible method for assessing patients’ response to IC, because radiological techniques are not always satisfactory [78,87]. Thirdly, RC enables the eradication of the residual disease and achieves a complete response in patients with partial remission and patients with an erroneous finding of complete response [87,88,89]. Finally, approximately three out of four patients who initially responded well to IC will experience a relapse at the site of the response [90]. 

Taking into consideration cN+ patients, the utility of lymphadenectomy remains a controversial topic. Currently, the standard treatment for patients with MIBC is RC with PLND with neoadjuvant chemotherapy [6]. As already stated, LNI is one of the most important prognostic factors, next to the local advancement, in BCa patients [11,91]. Consequently, PLND is inextricably linked with RC, as imaging techniques are poor nodal staging tools [92]. However, the therapeutic effect of PLND remains debatable. The available publications emphasize the oncological benefits of PLND during RC compared to its absence [93,94,95,96,97]. As of now, the ePLND remains the gold standard template. Several studies have been conducted on the oncological outcomes of the super-extended template, though none of them showed benefits in RFS, DSS, or OS [98,99,100]. A thorough presentation of the ePLND and sePLND extents with specific LN groups is illustrated in Figure 1. Figure 2 illustrates the anatomical compartments of PLND performed during the surgical procedure.

Lymph node metastases are detected in 16.7–29.3% of patients treated with PLND, which is associated with worse long-term oncological results [13,93,97,101,102]. Depending on the exclusion criteria, the number of patients in the study, stage of the tumor, method of diagnosis, and the chosen treatment method, lymph node metastases (pN+) were observed in 12.6% to 79.6% of patients with cN+, and even up to 91% when focusing on a specific group of patients [6,72,76,87,103]. An overview of the results is demonstrated in Table 3. This disproportion may be due to the use of NAC or AC. Based on a study of 3241 cN+ patients, Darwish et al. observed that treatment with NAC was associated with a significantly higher rate of downstaging to pN0 in comparison to surgical treatment alone (40.0% vs. 8.8%, OR = 6.88, *p* < 0.0001) [6]. The authors revealed that up to 91% of cN+ patients treated with RC without any form of ChT were pN+. In that cohort, patients who received NAC were pN+ in 60% of the cases. It was demonstrated that correct response to NAC and downstaging from cN1 to pN0 is associated with survival outcomes which are improved by up to 44% [104]. In a multicenter, retrospective study by Necchi et al., authors analyzed the outcomes of post-IC RC with PLND (n = 242) versus observation after IC (n = 280) in 522 cN+ patients [105]. It resulted in non-statistically significant improvement in OS for the post-IC surgery group (HR: 0.86, 95% CI: 0.56–1.31, *p* = 0.479). In another study, Al-Alao et al. revealed a poor OS, with a 5-year OS of 34% in cN+ patients treated with IC and RC [106]. Additionally, the authors observed heterogeneity in survival, ranging from 10% to 59% within 5 years, and proposed a risk-stratification tool. The study by Pak et al. showed incoherent results in different cN+ groups [107]. In the NAC group, the 5-year CSS of cN1-2 patients was improved compared to the RC group (68.1% vs. 52.9%; *p* = 0.035). Nevertheless, the 5-year CSS rate of cN3 patients was lower in the IC group than in the RC group (19.2% vs. 44.5%; *p* = 0.015). This study once again points out the importance of proper patient selection. Furthermore, a multitude of studies revealed improved oncological outcomes of PLND, although most of them pointed to considerably higher efficacy of a chemotherapy–surgery combination [82,108,109,110,111]. 

Nevertheless, several researchers have demonstrated promising results of PLND in cN+ patients, especially when placed into a proper clinical context. For example, it has been proven that removing more nodes can improve OS [11,112,113,114,115]. The researchers agreed that survival improvement positively correlates with the number of removed LNs, and this trend was independent of patients’ nodal status. Yet, it was reported that in order to achieve optimal oncological outcomes, a proper template of PLND is more important than focusing only on the total LN count [116]. However, it is essential to remember that some researchers demonstrated only 12.6% of cN+ patients to be truly N+ in post-PLND histopathological examination (pN+); therefore, the therapeutic value of PLND remains a subject of debate.

Although the therapeutic value of PLND might not be conclusive, the diagnostic and prognostic values are clear. Histological evaluation of the PLND specimen provides crucial information for further management. The number of LNs with metastases is an excellent indicator of the extent of the disease. Various factors have been reported as prognostic factors of BCa. Nevertheless, the pN status, next to the pT stage, is paramount. The increasing number of positive nodes is reflected in a worse prognosis for the patients. It was demonstrated that the median 3-year survival in patients with pN+ was 58.6%, 31.8%, and 6.8%, respectively, for one, two, five, and more positive LNs [3]. Other researchers obtained a similar correlation utilizing cutoff values of four, five, and six positive LNs [112,117,118]. Bruins et al., in an analysis of 369 pN+ patients, demonstrated better results in patients with a maximum of two positive LNs, achieving a 5-year relapse-free survival rate of 44% vs. 24% in the group with more than two positive LNs [119]. If the number of positive LNs is within a range of 1 to 4, the OS worsens with each additional metastatic LN. On the other hand, any positive LN after five does not alter the clinical outcome, because the mass of metastases is so significant. With such an unfavorable outcome of pN+ disease, it is recommended to treat every pN+ patient who did not undergo NAC with AC [18,22]. Therefore, information obtained from PLND can be utilized not only for prognosis and recurrence risk stratification but also to indicate the need for subsequent treatment. Another prognostic factor obtained from PLND is extracapsular invasion—microscopic perforations of LN capsules by neoplastic cells, which is associated with higher aggressiveness of cancer and poorer OS [118,120]. The diagnostic information obtained from resected LNs is essential and, for now, cannot be replaced by any other method. 

### 4.3. Neoadjuvant Chemotherapy (NAC)

ChT given before RC, as part of a multimodal approach, is recommended by most guidelines for all eligible MIBC patients, as well as in selected patients with moderate or high-risk NMIBC [19,21,121]. While the optimal specific regimen has not yet been established, the utilization of cisplatin-based NAC is now considered the gold standard, based on multiple studies confirming its major impact on OS in patients with BCa [122]. In 2016, Yin et al. performed a systematic review and meta-analysis which pooled 3285 patients from 15 randomized clinical trials and 13 retrospective studies, demonstrating a significant OS benefit (HR: 0.87, 95% CI: 0.79–0.96) [123]. A more recent study by Hermans et al. examined a larger group of patients (5517) and showed even greater benefits of NAC in BCa patients, particularly in the cT3-4a group (HR: 0.67, 95% CI: 0.51–0.89). In the cT2 group, the OS improvement was not very significant (HR: 0.91, 95% CI: 0.72–1.15) [124]. Furthermore, this improvement was achieved without noticeably affecting surgical morbidity [10]. However, the NAC effectiveness is not as clear as it might seem, and there are large discrepancies in the results. In another recent meta-analysis, Li et al. demonstrated similar OS in patients treated with NAC + RC versus RC alone (HR: 0.92, 95% CI: 0.84–1.00, *p* = 0.056) [125]. On the other hand, in 2020, Hamid et al., in a meta-analysis of 13,391 patients, addressed the former results and demonstrated an unequivocally positive effect of NAC on OS (HR: 0.82, 95% CI: 0.71–0.95, *p* = 0.009) [126].

The side effects of cisplatin, including nephrotoxicity, neurotoxicity, and decreased heart function, preclude 30–50% of BCa patients from safe cisplatin-based treatment [127]. The ineligibility criteria are summarized in Box 1. Various non-cisplatin-based alternatives, such as gemcitabine/carboplatin and immunotherapy with pembrolizumab or atezolizumab, have shown promising results, but there is insufficient high-level evidence to support their recommendation [128,129]. According to the EAU guidelines, NAC is recommended only for patients eligible for cisplatin-based ChT [19]. For ineligible patients, it is reasonable to consider a referral to a clinical trial [121].

Box 1BCa patients ineligible for cisplatin-based ChT [127].
WHO or ECOG performance status of 2, or Karnofsky performance status of 60–70%Creatinine clearance (calculated or measured) less than 1 mL/sCTCAE version 4, grade 2 or above audiometric hearing lossCTCAE version 4, grade 2 or above peripheral neuropathyNYHA class III heart failure


Miscellaneous combinations of cisplatin-based regimens exist: methotrexate, vinblastine, doxorubicin, and cisplatin (MVAC), gemcitabine and cisplatin/carboplatin (GC) being the most widely used for young and old patients (due to its less toxic profile), respectively. Alternatives include dose dense MVAC (DDMVAC), cisplatin and methotrexate (CM), cisplatin and 5-fluorouracil (5-FU), and cisplatin, methotrexate, and vinblastine (CMV) [123]. Numerous studies have been conducted comparing the effectiveness of different regimens; however, the results are inconsistent, and further research is needed to determine the best option definitively [130,131,132,133]. The previously cited meta-analysis by Yin et al. also examined this problem and compared the most popular regimens, showing similar pathological complete responses (pCR) of GC and MVAC, but a significantly reduced OS of GC (HR: 1.26, 95% CI: 1.01–1.57), which was probably influenced by the older age of GC patients [123]. An additional issue is a lack of consensus regarding the number of cycles to be administered, with most regimens recommending four cycles, but other options mentioned as well, which further hinders the comparison of the results [134].

The pCR to NAC appears to be one of the critical parts in predicting survival in MIBC patients. In a meta-analysis that pooled 886 patients from 13 trials, Petrelli et al. reported that patients who achieved pCR presented a relative risk for OS of 0.45 (95% CI: 0.36–0.56, *p* < 0.0001) [135]. This factor seems to be even more important in patients with cN+ BCa. The decision to continue further (including surgical) treatment depends on the response to ChT (IC) [87]. The reason is the poor prognosis of patients with residual pathologic nodal disease after ChT, contrasting with the relatively good outcomes of patients who achieved pCR [108,136]. Different studies have reported its significant benefit for cN+ patients, especially those achieving the pN0 category followed by consolidative surgery while initially presenting with node-positive disease, with one study reporting a 66% cancer-specific survival rate [78,87,137,138]. Patients with pCR after IC who did not undergo consolidative surgery are at a high risk of relapse; therefore, the surgery should not be spared [79,139]. On the other hand, most patients with weak or no response to IC will not benefit from consolidative surgery, with a very poor prognosis regardless of the treatment undertaken [78,87]. A study by Ploussard et al. compared OS outcomes in 450 N+ BCa patients at the time of RC, according to the ChT response. The authors revealed a significant association between the persistence of bladder invasion in RC specimens and OS, with an enormous HR of 2.40 (95% CI: 1.06–5.44) for those patients [140]. This demonstrates that the post-IC nodal status is critical, as it allows for an appropriate selection of patients for surgery.

### 4.4. Adjuvant Chemotherapy (AC)

The role of AC with RC in the treatment of cN+ BCa has not been fully established. Indicated benefits of this approach are that it allows immediate surgical treatment and proper pathological staging. There is still a lack of evidence from well-designed randomized phase III trials. With regard to cN+ BCa, another difficulty is that in many trials, inclusion criteria are not focused on cN+, but involve pT3/4 tumor stage and/or pN+ status. Based on a meta-analysis of nine randomized control trials, the utilization of immediate postoperative cisplatin-based AC resulted in an improvement of the OS. Nonetheless, a statistical significance level of this observation was borderline (*p* = 0.049) [141]. In available trials, authors used the following AC regimens: monotherapy with cisplatin, GC, CMV, cisplatin, cyclophosphamide, adriamycin (CISCA), MVAC, and CM [141,142,143,144,145,146]. 

Sternberg et al., in the randomized clinical trial, evaluated immediate AC versus deferred ChT at relapse after RC in 284 pT3/pT4 or N+ patients [147]. In their study, four cycles of immediate AC regimen and six cycles of deferred ChT regimen with GC, MVAC, or DDMVAC were used. The improvement in OS in patients with immediate AC was insignificant, but the authors emphasized that their study was limited in power. Therefore, it is believed that particular groups of patients might still benefit from immediate AC, and, for this purpose, a large meta-analysis with updated individual patient data is required. In another randomized clinical trial of 194 patients, Cognetti et al. evaluated the benefit of GC AC after RC versus RC alone [148]. Focusing on N+ patients, there were no differences between the mentioned groups in a 5-year DFS. This parameter, in AC patients, reached 18.9% compared to 19.4% in the RC group (*p* = 0.80). It should be noted that the performed clinical trials had some methodological flaws, which is why all results should be carefully analyzed. With the low statistical significance of the prospective trials, it is mandatory to discuss the outcomes of retrospective ones. In the multicenter study, Svatek et al. identified 3947 patients with BCa treated with RC without NAC, of whom 932 (23.6%) received AC [149]. The treatment with AC was independently associated with OS benefit (HR: 0.83, 95% CI: 0.72–0.97, *p* = 0.017). In this analysis, OS improvement was demonstrated, especially in N+ and advanced pathologic stage patients. Furthermore, Galsky et al., in another retrospective study of 5653 patients diagnosed with pT3-4 or pN+ BCa, compared the effectiveness of RC with that of RC plus AC. Their analysis showed improvement of OS in the group receiving AC (HR: 0.70; 95% CI: 0.64–0.76) [150]. Finally, Berg et al. retrospectively enrolled 15,397 patients who underwent RC (without NAC) and were diagnosed with T2N+ or ≥ T3N0/N+ [151]. The patients were identified in the National Cancer Database. The authors analyzed the impact of AC on OS regarding patients’ variant histology. In N+ patients, OS benefit was observed in pure urothelial carcinoma (HR: 0.87; 95% CI: 0.82–0.91), while no differences were reported in patients with other histological variants. In the urothelial carcinoma group, median OS values were 17.49 (95% CI: 16.79–18.07) and 26.78 (95% CI: 25.34–28.17), respectively, for all patients treated with RC as well as those with an addition of AC. Moreover, several studies reported that AC administration was associated with survival benefits in a group of N+ patients [152,153,154,155,156,157]. A recent report by Afferi et al. indicated that patients with more than three metastatic nodes are the group that will benefit from cisplatin-based AC after RC [158]. However, the most important issue with the aforementioned promising results of AC is the lack of trials distinguishing cN+ patients in their cohort.

Another question to consider is AC after NAC. There are limited data on this topic, and only retrospective data are available. RFS and DSS were reported after such management [159]. Reports are indicating that in N+ or pT3/T4 previously treated with NAC, AC might be associated with better OS [160].

### 4.5. Immunotherapy 

NAC is now standard in treating eligible patients with muscle-invasive urothelial carcinoma. The utilization of NAC increased from 9.7% in 2006 to 32.2% in 2014. However, there are patients ineligible for classical chemotherapy [161]. Factors influencing the use of NAC are higher comorbidity score, older age, disease-related impairment of renal function, poor performance status, presence of comorbidities that may be exacerbated by treatment-related toxicity, lower cT stage, patient poverty, and having undergone partial cystectomy [161,162,163]. These patients may benefit from the new neoadjuvant and adjuvant treatment modalities.

One of the new lines of therapy for patients is the treatment with pembrolizumab. Pembrolizumab is a potent monoclonal antibody of humanized immunoglobulin G4. It binds to PD-1 and inhibits the interaction with PD-L1 and PD-L2 ligands (programmed death-ligands) on tumor cells, thus blocking the PD-1/PD-L1 pathway and preventing T-cell inactivation [164]. Phase III KEYNOTE-045 results demonstrated the OS benefit of pembrolizumab in all subgroups as second-line therapy in patients with locally advanced and unresectable or metastatic BCa, including liver metastases and visceral metastasis that has progressed after platinum-based ChT. The median OS was 10.1 months (95% CI: 8.0–12.3) for pembrolizumab and 7.3 months (95% CI: 6.1–8.1) for chemotherapy. Additionally, median progression-free survival was 2.1 months (95% CI: 2.0–2.2) for pembrolizumab and 3.3 months (95% CI: 2.4–3.6) for chemotherapy. Median 1- and 2-year OS rates were higher with pembrolizumab than with chemotherapy (1-year OS: 44.2% vs. 29.8% and 2-year OS: 26.9% vs. 14.3%, respectively) [165,166]. The KEYNOTE-052 study demonstrated the efficacy and safety of first-line pembrolizumab therapy in cisplatin-ineligible patients with locally advanced and unresectable or metastatic bladder cancer [167,168,169]. Prolonged OS was observed, with an objective response rate (ORR) of 28.6% (95% CI: 24.1–33.5) [167]. An improvement in OS was reported, especially in patients with PD-L1 expression and lymph node-only disease. Pembrolizumab is currently approved in locally advanced or metastatic BCa patients who do not qualify for cisplatin treatment. Additionally, it can be utilized in patients with advanced or metastatic BCa who are progressing during or after platinum-containing chemotherapy, or within 12 months of platinum-based NAC or AC. Treatment is also approved in patients with the following: bacillus Calmette-Guerin (BCG) unresponsive BCa, high-risk BCa, and NMIBC with carcinoma in situ, (CIS) with or without papillary tumors. These patients are ineligible for RC or have not settled on undergoing surgery [19,164,170]. In the phase III trial MK-3475 AMBASSADOR, researchers investigated whether post-RC pembrolizumab would improve OS and DFS in patients with high-risk MIBC [171]. The trial outcomes may enable the utilization of pembrolizumab as an alternative to ChT adjuvant treatment. 

Atezolizumab is another PD-1/PD-L1 immune checkpoint inhibitor which the FDA has approved for the treatment of patients with metastatic or locally advanced urothelial carcinoma, whose disease progressed during or following platinum-containing ChT or within 12 months of NAC or AC platinum-containing treatment [172,173,174,175]. Results from the Phase 3 IMvigor211: 24-month OS rate was 23% with atezolizumab vs. 13% with ChT. Patients treated with ChT had more 3/4 grade TRAE than patients treated with atezolizumab: 43% vs. 22% [176]. The SAUL study assessed the effectiveness of the treatment in patients not eligible for the IMvigor211 phase 3 trial. In this study, median OS was 8.7 months (95% CI: 7.8–9.9), the 6-month OS was 60% (95% CI: 57–63%), the median PFS was 2.2 months (95% CI: 2.1–2.4), and the ORR was 13% (95% CI: 11–16%) [177]. The phase III IMvigor010 study was the largest and first-completed phase 3 trial to evaluate the role of a checkpoint inhibitor in the adjuvant therapy of MIBC [178]. However, the study did not meet the primary endpoint of improvement in DFS in the atezolizumab group compared to observation, and was terminated. Therefore, the results of new studies with different checkpoint inhibitors are awaited, which would allow the establishment of the position of immunotherapy in the adjuvant treatment of BCa. 

Results from JAVELIN Bladder 100 proved that maintenance treatment with avelumab (anti-PD-L1 antibody) significantly improves overall survival: 21.4 months (from the start of checkpoint inhibitor administration) in patients with advanced or metastatic urothelial carcinoma that has not progressed on 1 L platinum-containing ChT. Avelumab 1 L maintenance is approved as a level 1 evidence treatment in a particular group of patients [179,180].

The FDA has approved Erdafitinib for the treatment of locally advanced or metastatic BCa, which progresses on platinum-based ChT and has fibroblast growth factor receptor (FGFR) 3 or FGFR2 alterations. It is a tyrosine kinase inhibitor of FGFR1-4 that binds to receptors, blocks FGF’s activity and leads to cell death [21,181,182,183]. FGFR changes are present in 15–20% of metastatic BCa patients. Previous studies have shown an ORR of 40% (95% CI: 31–50%, including 3% complete response). However, erdafitinib exhibits ocular toxicity that calls for special attention [184,185,186]. The long-term follow-up of the phase II study showed a similar safety profile to the first analysis. Grade 3-4 TRAE occurred in 72/101 enrolled patients, but there were no treatment-related deaths in the follow-up analysis [187].

New research is emerging to develop drugs that can be combined with PD-1/PD-L1 inhibitors or administered interchangeably. Enfortumab vedotin was created by combining an antibody and a drug. The antibody is directed against nectin-4; the drug leads to disruption of the microtubules. This causes a cell cycle arrest in nectin-4 expressing cells [188,189]. Enfortumab vedotin, in the first phase study, (EV-101 NCT02091999) demonstrated safety, tolerability, and antitumor activity in patients with Nectin-4-positive solid tumors who progressed on a ≥1 prior chemotherapy regimen and/or anti-PD-1/L1 [190,191,192]. Phase II study results show that the drug is effective: its overall response rate (ORR) was up to 52%; its duration of response (DOR) was 7.6 months (95% CI: 4.93–7.46); its OS was 11.7 months (95% CI: 9.1, not reached), and the drug was safe. The most common treatment-related adverse events (TRAEs) were peripheral neuropathy, rash, decreased appetite, fatigue, dysgeusia, and alopecia [188,192]. Enfortumab vedotin is utilized in the treatment of patients with locally advanced or metastatic urothelial cancer, who have previously received a PD-1 or PD-L1 inhibitor and platinum-containing NAC or AC [189].

In 2021 FDA issued expedited approval for the utilization of sacituzumab govitecan in metastatic BCa or locally advanced patients who have previously received platinum-based ChT and a PD-1/PD-L1 inhibitor. Sacituzumab govitecan is an antibody–drug conjugate consisting of an active metabolite of irinotecan and Trop-2 directed anti-Trop-2 checkpoint inhibitors. A phase II study (TROPHY-U-01) has shown the benefits of this drug. ORR was 27.4% (95% CI: 19.6–36.9), median DOR was 7.2 months (95% CI: 4.7–8.6), median PFS was 5.4 months (95% CI: 3.5–7.2), and OS was 10.9 months (95% CI: 9.0–13.8) [193,194,195].

### 4.6. Future Perspectives

Research is currently being carried out on new molecules and a new application of the current drugs. Phase III trials are currently underway, with perioperative pembrolizumab monotherapy or combined with enfortumab vedotin, and RC plus PLND versus RC plus PLND alone in cisplatin-ineligible patients with MIBC (KEYNOTE-905/EV-303). Additionally, in the phase III trial KEYNOTE-866, researchers will check the effectiveness of neoadjuvant chemotherapy with either perioperative pembrolizumab or placebo in previously untreated cisplatin-eligible patients with MIBC [164].

There are also clinical trials on the combination of pembrolizumab with enfortumab vedotin in treating patients with cisplatin-ineligible locally advanced or metastatic BCa. The results of the conducted studies confirm the safety of the treatment. In addition, the ORR was 73.3% (95% CI: 58.1–85.4), 12-month DOR was 53.7% (95% CI: 27.4–74.1), 12-month OS was 81.6% (95% CI: 62.0–91.8) [196,197]. Currently, phase II trials are underway using durvalumab (PD-L1 inhibitor) and tremelimumab (CTLA-4 inhibitor) as a neoadjuvant treatment in patients with MIBC. It has been found to be safe and active in patients with MIBC regardless of tumor immune score [198]. Phase 2 trials also confirm the antitumor effect of camrelizumab (PD-1 inhibitor) with famitinib in patients with advanced or metastatic BCa who had progressed after platinum-based ChT. The subgroup of BCa patients achieved a median PFS of 8.3 months (95% CI: 4.1–not reached), and an ORR of 38.9% (95% CI: 17.3–64.3%) [199]. Famitinib malate is a tyrosine kinase inhibitor (TKI) against VEGFR-2, PDGFR, c-kit, and FGFR [200]. The phase I NABUCCO study showed the effectiveness of neoadjuvant therapy with ipilimumab (CTLA-4 inhibitor) and nivolumab (PD-1 inhibitor). In patients with stage III BCa treated with this combination, resection was possible within 12 weeks of starting therapy in 23 patients (96%) [201]. The phase III CheckMate 274 study compared nivolumab with placebo in the adjuvant setting in patients with muscle-invasive urothelial carcinoma [202]. In the study on patients with a high risk of MIBC who underwent major surgery, DFS was longer in the adjuvant nivolumab group than in the placebo group of patients, with PD-L1 and PD-L1 expression levels of 1% or more. The median DFS in the nivolumab-treated population was 20.8 months (95% CI: 16.5–27.6), and 10.8 months (95% CI: 8.3–13.9) for placebo. The percentage of patients who were alive and disease-free at six months was 74.9% with nivolumab vs. 60.3% with placebo. In August 2021, the FDA approved nivolumab as an adjuvant treatment for BCa patients with a high risk of recurrence after RC [203]. CheckMate 275 has been certified with the durable antitumor activity of nivolumab [204]. An overview of currently conducted clinical trials is demonstrated in Table 4. 

The results of these studies will introduce new guidelines for treating advanced, N+, or ChT-ineligible patients with BCa. The current management of cN+ patients is simplified in Figure 3. 

## 5. Conclusions

The management of patients with cN+ BCa remains imprecise in many aspects. From diagnostics to surgical treatment and ending with systemic treatment, high-value clinical research is lacking. However, the available data allow for some important statements. Multimodal treatment with NAC and RC achieves the best prognosis for patients with cN+ BCa. Emerging evidence indicates that AC results are comparable to NAC; however, there is still a lack of definitive research. Nevertheless, the response to ChT is crucial as a prognostic factor for cN+ patients, and AC should be administered to the patients who have not received NAC. Due to the high percentage of ChT-ineligible BCa patients, neoadjuvant and adjuvant immunotherapy is gaining more and more importance in clinical practice. The results of many currently carried out clinical trials regarding immunotherapy may implement changes to the guidelines in the near future. In cN+ patients, if RC is performed, the PLND should not be omitted, and the extended template should be utilized to provide necessary diagnostic data. Moreover, the total resected node count is less important than the range of PLND. 

## Figures and Tables

**Figure 1 cancers-14-05286-f001:**
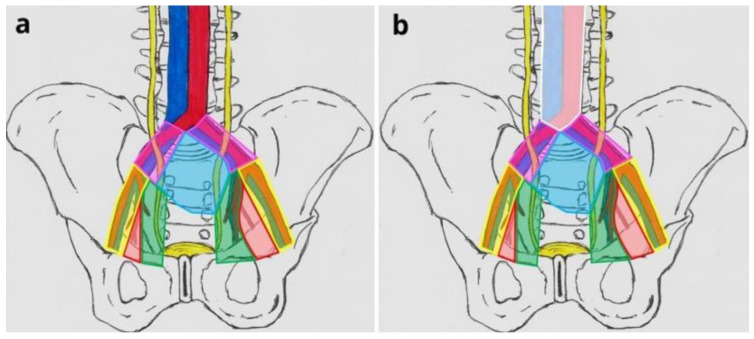
The anatomical diagram of PLND divided into templates: (**a**)—extended, and (**b**)—super-extended; the obturator fossa (red), external iliac vessels (yellow), internal iliac vessels (green), common iliac vessels (pink), the presacral area (blue), and the paraaortic/paracaval area (white).

**Figure 2 cancers-14-05286-f002:**
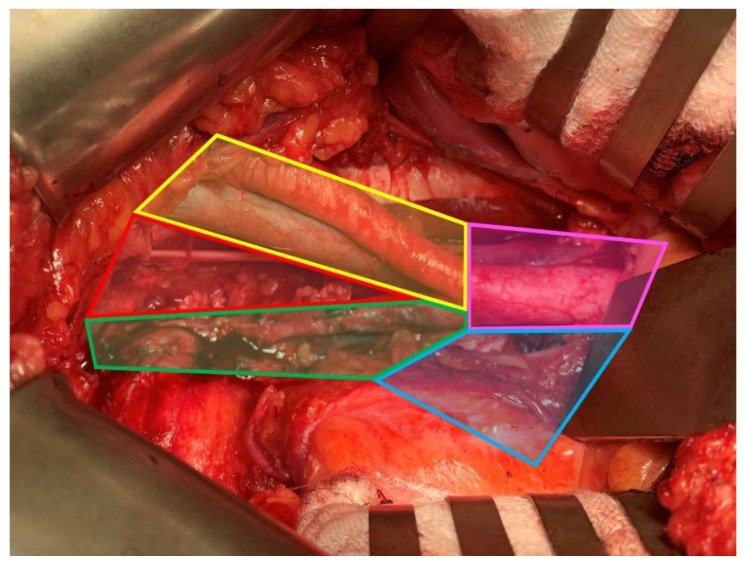
Superimposing of the anatomical areas of PLND during RC; the obturator fossa (red), external iliac vessels (yellow), internal iliac vessels (green), common iliac vessels (pink), and the presacral area (blue).

**Figure 3 cancers-14-05286-f003:**
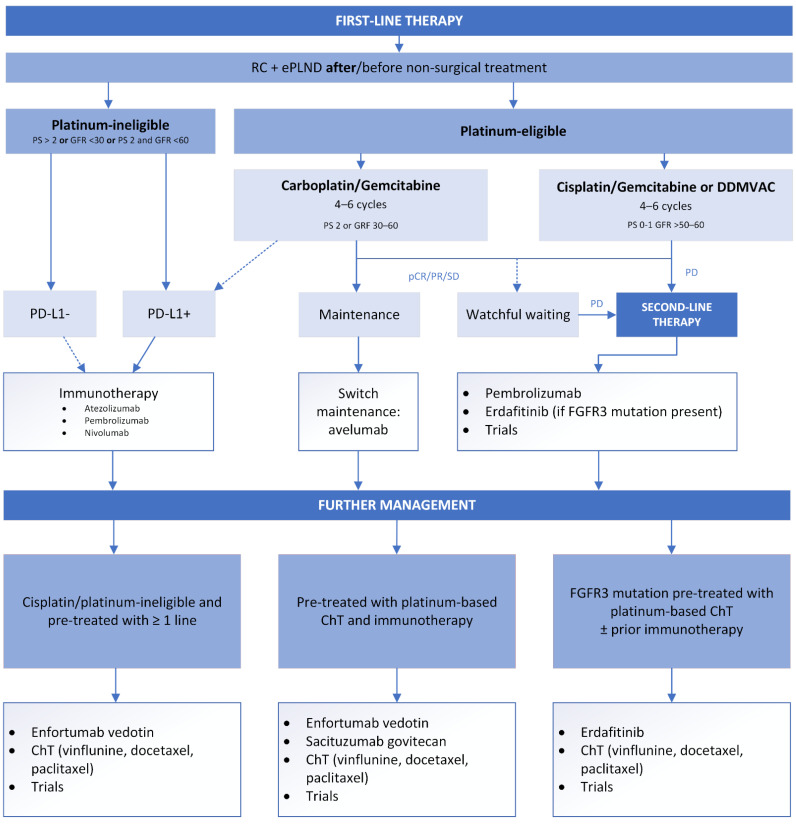
Summary of the management of cN+ patients. FGFR = fibroblast growth factor receptor; GFR = glomerular filtration rate; pCR = pathological complete response; PD = progressive disease; PD-L1 = programmed death-ligand 1; PR = partial response; PS = performance status; SD = stable disease.

**Table 1 cancers-14-05286-t001:** Comparing CT, MRI, and PET/CT performance in preoperative prediction of LNI in BCa patients.

Study Authors	Year	n	Sensitivity (%)	Specificity (%)	Accuracy (%)
Computed Tomography (CT)					
Vock et al. [52]	1982	77	-	-	89
Buszello et al. [53]	1994	50	33	100	-
Paik et al. [54]	2000	82	19.1	96.7	-
Ficarra et al. [55]	2005	156	42.2	100	76.9
Baltaci et al. [56]	2008	100	30.7	94.3	86
Tritschler et al. [57]	2012	219	30.4	90	71.2
Magnetic resonance imaging (MRI)					
Buy et al. [58]	1988	40	83.3	100	-
Tavares et al. [59]	1990	29	50	100	82
Deserno et al. [38]	2004	58	96	95	95
Daneshmand et al. [34]	2012	122	40.7	91.5	80.3
Thoeny et al. [60]	2014	120	63–78	79–85	75–83
Wu et al. [61]	2018	103	44.8	93.2	79.6
Positron emission tomography (PET)/Computed Tomography (CT)					
Swinnen et al. [49]	2009	51	46	97	84
Maurer et al. [39]	2012	44	58	66	64
Brunocilla et al. [42]	2014	26	42	84	73
Soubra et al. [43]	2016	78	56	98	-
Pichler et al. [48]	2016	70	63.6	86.4	82.9
Moussa et al. [44]	2021	134	40.3	79.5	62

**Table 2 cancers-14-05286-t002:** Overview of cN+ patients’ management strategies according to guidelines provided by the EAU, the AUA, the ESMO, the NCCN, and the NICE.

Guidelines	Management Strategies
EAU	- NAC + RC - RC + AC (for patients who didn’t receive NAC) - Radical ChRT - Nivolumab immunotherapy (advised for pN+ patients not eligible for, or who declined, AC)
AUA	- NAC + RC - RC + AC (for patients who didn’t receive NAC)
ESMO	- NAC ± RC
NCCN	For N1 patients: - NAC + RC (especially for cN1 patients) - RC (for chemotherapy-disqualified patients) - Bladder preservation + ChRT - ChRT - RT For N2-3 patients: - Downstaging ChT - ChRT
NICE	- NAC + RC - RC + AC

EAU: The European Association of Urology; AUA: The American Urological Association; ESMO: The European Society for Medical Oncology; NCCN: The National Comprehensive Cancer Network; NICE: National Institute for Health and Care Excellence; NAC: neoadjuvant chemotherapy; RC: radical cystectomy; AC: adjuvant chemotherapy; ChT: chemotherapy; ChRT: chemoradiotherapy; RT: radiotherapy.

**Table 3 cancers-14-05286-t003:** The comparison of studies presenting the percentage of pN+ in cN+ patients.

Study Authors	Year	cN+	pN+	pN0	% of pN+
Moschini M et al. [72]	2020	221	28	193	12.7%
Herr H et al. [106]	2004	1091	216	875	19.8%
Zargar-Shoshtari et al. [76]	2015	cN1 = 133 cN2 = 134 cN3 = 15 cN+ = 282	59 68 8 135	74 66 7 147	44.4% 50.7% 53.3% 47.8%
Ho et al. [88]	2016	55	25	30	45.5%
Darwish et al. [6]	2020	3241	1286 *	330 *	79.6%

* Missing data of pN+ in 1625 patients; cN+—clinically positive lymph nodes; cN1—clinical metastasis in a single lymph node in the true pelvis (hypogastric, obturator, external iliac, or presacral); cN2—clinical metastasis in multiple regional lymph nodes in the true pelvis (hypogastric, obturator, external iliac, or presacral); cN3—clinical metastasis in a common iliac lymph node(s); pN+—pathologically positive lymph nodes; pN0—no nodal metastases.

**Table 4 cancers-14-05286-t004:** Currently ongoing clinical trials on immunotherapy in BCa.

Name of Clinical Trial	Phase	Drug	Recruitment Status on 22 July 2022	Number of Participants	Participants with:
MK-3475-045/KEYNOTE-045 (NCT02256436) [165,205]	III	Pembrolizumab	Completed	542	metastatic or locally advanced/unresectable BCa with recurrence or progression after platinum-based ChT.
KEYNOTE-052 (NCT02335424) [206]	II	Pembrolizumab	Completed	374	metastatic or locally advanced/unresectable BCa ineligible for cisplatin-based ChT.
EV-101 (NCT02091999) [207]	I	Enfortumab vedotin	Active, not recruiting	155 (BCa)	nectin-4-positive BCa/other solid tumors, with progression or ineligible for platinum-based ChT and/or anti-PD-1/L1 therapy.
EV-201 (NCT03219333) [208]	II	Enfortumab vedotin	Active, not recruiting	125	cisplatin ineligible metastatic or locally advanced BCa who progress on/after PD-1/L1 inhibitors.
EV-301 (NCT03474107) [209]	III	Enfortumab vedotin	Active, not recruiting	608	metastatic or locally advanced BCa with recurrence or progression after PD-1/PD-L1 inhibitors.
IMvigor211 (NCT02302807) [210]	III	Atezolizumab	Completed	931	metastatic or locally advanced BCa with progression during/after platinum-based ChT.
SAUL (NCT02928406) [177,211]	III	Atezolizumab	Active, not recruiting	1004	metastatic or locally advanced/unresectable BCa with progression during/after one to three prior therapies.
IMvigor010 (NCT02450331) [178,212]	III	Atezolizumab	Completed	809	pT3-T4a or pN+ MIBC.
JNJ-42756493 (NCT02365597) [185,213]	II	Erdafitinib	Recruiting	236	metastatic or unresectable BCa that harbor specific FGFR genomic alterations.
JAVELIN Bladder 100 (NCT02603432) [214]	III	Avelumab	Active, not recruiting	700	metastatic or locally advanced/unresectable BCa without progression after first-line ChT.
TROPHY-U-01 (NCT03547973) [215]	II	Sacituzumab govitecan	Recruiting	321	metastatic BCa unresponsive to platinum-based ChT or PD-1/PD-L1 inhibitors.
KEYNOTE-905/EV-303 (NCT03924895) [216]	III	Pembrolizumab + Enfortumab vedotin + RC + PLND	Recruiting	857	MIBC who are cisplatin-ineligible or decline ChT.
KEYNOTE-866 (NCT03924856) [217]	III	Pembrolizumab	Recruiting	870	MIBC who are cisplatin-eligible.
AMBASSADOR (NCT03244384) [171,218]	III	Pembrolizumab	Active, not recruiting	739	locally advanced BCa or MIBC.
EV-103/KEYNOTE-869 (NCT03288545) [197]	I/II	Enfortumab vedotin + Pembrolizumab	Recruiting	457	metastatic or locally advanced BCa who are cisplatin-ineligible.
DUTRENEO (NCT03472274) [219]	II	Durvalumab + Tremelimumab	Active, not recruiting	99	cT2-T4N0-1M0 BCa who are cisplatin-eligible, candidates to RC.
SHR-1210 (NCT03827837) [220]	II	Camrelizumab + Famitinib	Recruiting	265	unresectable BCa after failure of ≤2 platinum-based ChT.
CheckMate 274 (NCT02632409) [202,221]	III	Nivolumab	Active, not recruiting	709	invasive urothelial cancer at high risk of recurrence after RC.
CheckMate 275 (NCT02387996) [222]	II	Nivolumab	Completed	270	metastatic or locally advanced/unresectable BCa with recurrence or progression after platinum-based ChT.

## Data Availability

Not applicable.

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
