# Peer review of "Management of Bladder Cancer Patients with Clinical Evidence of Lymph Node Invasion (cN+)"

_cancers, 2022, doi:10.3390/cancers14215286_

Round 1

Reviewer 1 Report

In this review, the authors seek to summarize the evidence surrounding the management of patients with clinically node positive bladder cancer. They have performed a very extensive review of the literature that should be commended, but this manuscript is rather broad and I have some suggestions that may improve the message.

-       First, I recommend further editing as there are too many grammar/linguistic errors for publication in an English language journal.

-       Second, I recommend streamlining the manuscript some more as it covers too many topics. The major flaw is that you don’t truly focus on cN+ patients, instead discussing the management of pN+ patients and systemic therapies for metastatic disease. We should hear the authors summary of the literature and expert opinion on how patients should be managed when they present with clinically enlarged lymph nodes.

Abstract:

-       The statement that there is lack of evidence of the superiority of neoadjuvant therapy over adjuvant therapy is false. All guidelines support the use of neoadjuvant chemotherapy in this population, often termed induction chemotherapy. It is true that cN+ patients have not been included in trials, but all data supports the use of NAC in high risk groups. Saying otherwise is against expert consensus.

-       It is true that recent studies support the use of immunotherapy, but relevant to this topic is the adjuvant immunotherapy data that you do not summarize i.e. checkmate 274, imvigor 010 and ambassador

Introduction:

-       Where do you get the data that 5 year OS is only 39-56% in node negative patients? It is much higher than this, especially in responders to neoadjuvant chemo.

-       You should have a methods section stating what type of review you performed and search strings

Section 2:

-       Focus only on the ability of imaging to determine nodal status before surgery. Mentions of chest imaging and retrograde urogropahy is superfluous for this discussion.

-       Table 1 should say computed tomography not computer tomography. Would also include PET in this table.

-       The mention of nomograms is very superficial, but this could be interesting for the manuscript. Mention what goes into the nomograms etc.

Section 3:

-       This section is somewhat off topic as the debate over limited and extended templates applies more to clinically node negative patients. Node positive patients should have a thorough and extended dissection. This is widely considered standard, and you should give us your expert insight on this rather than reporting all the various studies that do not apply in this special scenario. i.e. in Nechi’s study from 2019 talk about the important difference in outcomes with pelvic nodes and retroperitoneal nodes. In the LEA trial talk about the well known limitations of the study.

-       Note that super extended templates include paracaval and paraaortic nodes

Section 4:

-       Your explanation of the will rogers phenomenon is a bit confusing and again is more relevant when talking about cN0 patients who get a node dissection vs not or less extended vs more extended.

-       The summary of the guideline statements in section 4.1 are not entirely accurate

-       Section 4.2. I am not sure why you get into the discussion of node counts as you later acknowledge that the consensus is that anatomic template is most important. As you also showed in earlier section and in your figures.

-       Again, I don’t think a summary of the use of chemo/IO/targeted agents is within the scope of this review. Focus on cN+ patients. They should all get chemo followed by surgery and if they can’t get chemo up front, we have new data to support IO in an adjuvant fashion. Summarize those trials. We don’t really need to hear about the trials done in cN0 patients.

Reviewer 2 Report

Review is not the simple list of articles, the authors should have your own views and speculations. Moreover, the abstract was templating.

Reviewer 3 Report

The manuscript of Malkiewicz and colleagues reviews the available options for clinical management of lymph-node-positive bladder cancer The review is comprehensive and well written, with all my comments focusing on the enhancement of its curb appeal to readers.

1)    The Paper is very comprehensive, but long and perhaps suboptimally structured. Its readers need to review lots of material before getting to the summary/conclusion section (which is significantly clearer than the abstract). A better summary of the overall goal and unique focus of the review and a better summary of the key take home points in the abstract could strongly enhance its appeal of to the readers.  

2)    I suggest editing the review to reduce its overall size. Currently, a significant volume of narrative is dedicated to discussion of non-definitive studies, lack of established criteria etc. Addressing these controversies s important, ut can be done in a more concise manner to avoid diluting the more definitive points.

3)    The section on immunotherapy, introduced in the abstract as being “of growing importance” is currently underdeveloped. Introduction of additional narrative and a figure explaining the concepts underlining immunotherapy of bladder cancer and rationale for the discussed combinations with chemo- and targeted therapies could make the paper more novel and attractive.

4)    Current figure 1 does not work for this reviewer. Please consider a revision to make the differences between panels/options clearer and the massage more transparent.

5)    Tables 3 and 5 contain some empty spaces and can be significantly condensed.

6)    (minor): Page 16: I suggest rephrasing the reference to checkpoint inhibitors, to avoid describing them as immunoglobulin (suggesting polyclonal nature). Antibodies or just checkpoint inhibitors may be more precise. The section on immunotherapy deserves additional attantion.

Round 2

Reviewer 3 Report

The authors have addressed my key comments.